# Transcriptome Analysis of *Bacillus amyloliquefaciens* Reveals Fructose Addition Effects on Fengycin Synthesis

**DOI:** 10.3390/genes13060984

**Published:** 2022-05-31

**Authors:** Hedong Lu, Hai Xu, Panping Yang, Muhammad Bilal, Shaohui Zhu, Mengyuan Zhong, Li Zhao, Chengyuan Gu, Shuai Liu, Yuping Zhao, Chengxin Geng

**Affiliations:** 1School of Life Science and Food Engineering, Huaiyin Institute of Technology, Huaian 223003, China; luhd100@hyit.edu.cn (H.L.); sliverhai@163.com (H.X.); climbing123@163.cm (P.Y.); bilaluaf@hotmail.com (M.B.); shaohz2022@163.com (S.Z.); zhongmengyuan0829@163.com (M.Z.); twin-li@163.com (L.Z.); guchengyuan202108@163.com (C.G.); liushuai007@hyit.edu.cn (S.L.); zhaoyuping@hyit.edu.cn (Y.Z.); 2National Engineering Research Center for Functional Food, School of Food Science and Technology, Jiangnan University, Wuxi 214122, China

**Keywords:** fengycin, *Bacillus amyloliquefaciens*, transcriptomic, metabolism, regulation

## Abstract

Fengycin is a lipopeptide produced by *Bacillus* that has a strong inhibitory effect on filamentous fungi; however, its use is restricted due to poor production and low yield. Previous studies have shown that fengycin biosynthesis in *B**. amyloliquefaciens* was found to be significantly increased after fructose addition. This study investigated the effect of fructose on fengycin production and its regulation mechanism in *B**. amyloliquefaciens* by transcriptome sequencing. According to the RNA sequencing data, 458 genes were upregulated and 879 genes were downregulated. Transcriptome analysis results showed that fructose changed the transcription of amino acid synthesis, fatty acid metabolism, and energy metabolism; alterations in these metabolic pathways contribute to the synthesis of fengycin. In an MLF medium (modified Landy medium with fructose), the expression level of the fengycin operon was two-times higher than in an ML medium (modified Landy medium). After fructose was added to *B**. amyloliquefaciens*, the fengycin-synthesis-associated genes were activated in the process of fengycin synthesis.

## 1. Introduction

*Bacillus amyloliquefaciens* is a biocontrol strain that is commonly used to control pathogenic fungi and bacteria in plants. Fengycin is one of the amphipathic cyclic lipopeptide molecules produced by *B**. amyloliquefaciens* [1]. It can interact with the lipid bilayer of the cell membrane to change its structure and permeability, making it resistant to a variety of plant pathogens [2]. Due to its broad antibacterial range, safe degradation, and low hemolysis, Fengyin has the potential to be widely employed in agriculture, food, and medicine [3]. Fengycin comprises 10 amino acids and a C_14_-C_18_ fatty acid chain [4]. Fengycin is synthesized by nonribosomal peptide synthetase (NRPS), the operon encoding fengycin synthetase is about 38 kb in length, consisting of five open reading frames of *fenC*, *fenD*, *fenE*, *fenA*, and *fenB* [5]. Each module in the NRPS is responsible for adding monomeric amino acids to the extended peptide chain and catalyzes the substrate activation, covalent binding, and peptide bond formation steps in the fengycin synthesis process [6].

Fengycin possesses a wide range of biological functions, but its industrial applications are limited due to the low yield and high production costs. Biological metabolic modification can obtain high-producing lipopeptide strains, but it is crucial to understand what factors influence fengycin synthesis. An analysis of transcriptome after fructose treatment could find key genes involved in the biosynthesis of fengycin. The carbon source, nitrogen source, metal ion [7], light, temperature [8], and other factors have been demonstrated to affect the production of fengycin synthase in previous investigations. Fengycin production has been increased by changing the expression level of fengycin operon [9,10]. Moreover, the expression of metabolic genes in *Bacillus* influences fengycin production. Genes of *accA*, *cypC*, and *gapA* were found to be associated with the synthesis of fengycin [11]. The PhoR/PhoP two-component system regulates fengycin production in *B.subtilis* NCD-2 [12]. Adding different carbohydrates to the culture has been shown to influence the bacterial metabolic process in some investigations [13]. According to other studies, introducing fructose increases the bacterial breakdown of organic contaminants. The addition of fructose promotes the ability of the bacterial flora to degrade the dye [14]. Furthermore, fructose causes a high production of polyhydroxyalkanoate(PHA) in *Arctic Pseudomonas* sp. [15].

In our previous work, we demonstrated that fructose efficiently boosts the production of fengycin to 392.87 mg/L in *B. amyloliquefaciens* [16]. In this study, efforts were made to further reveal the fructose-induced metabolic mechanism of fengycin synthesis. The results offer gene expression information to further research on fengycin metabolic pathways.

## 2. Materials and Methods

### 2.1. B. amyloliquefaciens Strain and Culture Medium

The *B**. amyloliquefaciens* fmb-60 (CGMCC7.222) was cultured in 50 mL Landy medium (5 g/L sodium chloride, 5 g/L beef extract, 5 g/L yeast extract, 10 g/L peptone, 10 g/L glucose) at 37 °C and 160 rev min^−1^. Next, 5% culture solution was transferred to 100 mL modified Landy medium (ML medium, 1 g/L yeast extract, 14 g/L L-sodium glutamate, 0.5 g/L KCl, 1 g/L KH_2_PO_4_, 0.5 g/L MgSO_4_, 5 mg/L MnSO_4_, 0.17 mg/L CuSO_4_, 0.15 mg/L FeSO_4_) and modified Landy medium fructose addition (MLF medium, add 1.5 g fructose to 100 mL ML medium) incubated in a shaker for 24 h (33 °C and 160 rev min^−1^).

### 2.2. RNA Extraction and RNA Sequencing

RNeasy extraction kit (Invitrogen Life Technologies, Gaithersburg, MD, USA) was used to extract the total RNA of *B**. amyloliquefaciens* fmb-60. The concentration and purity of total RNA were determined by NanoDrop 2000 (Thermo Fisher Scientific, Waltham, MA, USA). The extracted total RNA was used as a template to synthesize the cDNA first strand using HiScript First Strand cDNA Synthesis Kit (Vazyme, Nanjing, China). The qualified double-stranded cDNA samples were sequenced by HiSeqTM 2000 platform (Illumina, San Diego, CA, USA) for high-throughput sequencing.

### 2.3. Transcriptome Data and Differential Gene Expression Analysis

RNA sequencing (RNA-Seq) was analyzed by the BGI Genomics Illumina platform (Shenzhen, China, http://www.genomics.cn/index.php (accessed on 5 September 2021). The low-quality reads and adapters were removed. Then, these clean reads were mapped to the reference genome (*B**. amyloliquefaciens* Y2) using a short oligonucleotide sequence alignment program (SOAP developed by Shenzhen Hua da Gene). Unigenes differentially expressed were screened using a false discovery rate (FDR) of <0.001 and log2 (fold change) of ≥1 as the criteria. The functional enrichment analysis of genes was conducted using Blast2Go (http://www.blast2go.org (accessed on 16 October 2021) software, and identified differentially expressed genes (DEGs) were used for Gene Ontology (GO) and Kyoto Encyclopedia of Genes and Genomes (KEGG) enrichment analysis. For gene ontology and KEGG pathway enrichment, Cluster performed (3.0) and Java Tree View software (1.1.6r2) were implemented, and enrichment results were filtered with the parameters of adjusted *p*-value < 0.05.

### 2.4. Quantitative Real-Time PCR (qRT-PCR)

The qRT-PCR reaction system is based on the instructions of the TaKaRa SYBR Premix ExTaqTMII kit (Takara, Kyoto, Japan). The primers required for the experiment are listed in Table 1. After adding primers and 1 ng cDNA template into the reaction system, the system was set up according to the following procedure: (1) 95 °C pre-denaturation for 5 s. (2) 95 °C denaturation for 5 s. (3) Extension at 60 °C for 30 s, and lasting for 40 cycles, and the signal collection step is also in the extension at 60 °C. (4) Denaturation at 95 °C for 15 s, then extension at 60 °C for 60 s, and denaturation at 95 °C for 15 s. The relative expression level of the target gene was analyzed three times.

## 3. Results

### 3.1. RNA-Seq Data Analysis

After transcriptome sequencing, the original MLF combination and ML group data were analyzed and summarized. The two databases generated about 13.3 and 13.6 million raw read data, respectively (Table 2). The total number of sequenced bases was 1197332460 and 1222740360, respectively, which can reach 100% coverage of the reference genome. After filtering raw reads, removing adaptors containing reads, and discarding low-quality reads, the number of reads was counted to compare different gene positions. As shown in Figure 1, reads are evenly distributed in each part of the gene. Through a global analysis of the clear reads data, the high-quality sequences obtained by sequencing are regraded to be relatively high, which can meet the needs of subsequent transcriptome analysis.

### 3.2. Differentially Expressed Genes GO Classification and KEGG Pathway

A total of 3617 differentially expressed genes were found by comparative analysis of transcriptomics data. Data samples with FDR ≤ 0.001 were screened, and the screening conditions were: a *p*-value less than 0.05 and gene fold change by more than two-times. A total of 1337 substantially differentially expressed genes were found after the screening. *B**. amyloliquefaciens* fmb-60 exhibited 458 genes upregulated and 879 genes downregulated in the MLF medium compared to the ML medium without fructose (Figure 2).

Differentially expressed genes discovered in *B**. amyloliquefaciens* fmb-60 were analyzed for GO enrichment, and the results were classified into three categories: biological processes, cellular components, and molecular functions (Figure 3). The expression of the *B**. amyloliquefaciens* fmb-60 gene is mainly concentrated in the transport process, metabolic activity, cell membrane creation, stress response, and biological regulation process, when comparing the ML and MLF groups.

KEGG cluster analysis of differentially expressed genes revealed that differentially expressed metabolic pathways are concentrated on the amino acid metabolism pathways (Figure 4), such as Histidine (ko00340), Arginine (ko00330), Alanine (ko00250), Glutamate (ko00470), Valine (ko00290). Bacterial chemotaxis (ko00230), phosphate transport system (ko00260), flagellar assembly (ko00240), and fatty acid metabolism (ko00071); all of these were altered.

### 3.3. Differentially Expressed Genes Analysis

RNA-seq results support the previous research, demonstrating that the fructose-induced condition of *B**. amyloliquefaciens* is a multigenic response. Several metabolic pathways change with fructose condition, which involves changes in metabolisms, such as amino acid metabolism, fatty acid metabolism, energy metabolism, transport system, and gene transcription and regulation (Figure 5). In addition, the transcriptome data revealed that the expression of the fengycin synthase operon had increased by two-times. As shown in Figure 6, changes in multiple metabolic pathways affect the synthesis of fengycin.

#### 3.3.1. Amino Acid Metabolism

According to the transcriptome data, the amino acid metabolism process undergwent significant alterations. Transcription of valine, leucine, isoleucine, and serine biosynthetic were upregulated, including branched-chain amino acid aminotransferase (*y**waA*), acetolactate synthase catalytic subunit (*ilvB*), and glycerate dehydrogenase (*serA*). The genes of alanine racemase (*y**nc*) and glutamate dehydrogenase (*rocG*) were downregulated.

#### 3.3.2. Fatty Acid Metabolism

Fatty acids are key components of fengycin, and the content of fatty acids in *B**. amyloliquefaciens* affects the biosynthesis of fengycin. Transcriptome sequencing results found that the gene expression involved in fatty acid metabolism changed, such as S-malonyltransferase (*bmyD*), bacilysin biosynthesis oxidoreductase (*ywfH*), 3-oxoacyl-acyl-carrier protein reductase during fatty acid biosynthesis (*DfnC*), and acyl-CoA dehydrogenase (*ydbM*) transcript content, with all being increased. Fatty-acyl-CoAsynthase (*yhfL*), long-chain fatty-acid-CoA ligase (*lcfA*), CoA dehydrogenase (*yngJ*), acyl-CoA dehydrogenase (*yusJ*), enoyl-CoA hydratase (*ysiB*) and other fatty-acid-degradation-pathway-related genes were down-regulated.

#### 3.3.3. Energy Metabolism

A total of 24 DEGs linked with energy metabolism were discovered through transcriptomic analysis. NADH dehydrogenase (*yqjM*), cytochrome c oxidase subunit assembly protein (*ctaA*), and cytochrome ubiquinol oxidase (*qoxABCD*) were upregulated. Electron transfer flavoprotein alpha subunit (*etfA*), electron transfer flavoprotein beta subunit (*etfB*), cytochrome c oxidase (*ctaDEFG*), ATP synthase (*yqjB*), and oxidative-phosphorylation-related genes were downregulated.

#### 3.3.4. Transport System

The transcriptional analysis revealed 81 DEGs associated with the transport system were identified under fructose conditions, including 13 in the phosphotransferase system (PTS), 29 in the ATP-binding cassette transporters (ABC transport system), 16 in the major facilitator superfamily transport system (MFS), and 23 other transport-related genes. Fructose transporter (*fruA*), glucose symporter (*glcP1*), multiple sugar transport system permease protein (*yurM*, *yurN*), amino acid transporter amino acid transporter, AAT family (*gabP*, *ybgF*), polar amino acid transport system (*ytmJ*, *ytmK*, *ytmL*, *ytmM*, *ytmN*), ammonium salt transporter (*narK*) chloramphenicol resistance protein (*ybcL*) and 26 other genes were upregulated. Lactose/L-arabinose transport system (*araP*, *araN*, *araQ*), ribose transport system (*rbsA*, *rbsB*, *rbsC*, *rbsD*), PTS system component (*mtlA*, *mtlA1*, *ywbA*, *licA*, *licB*, *licC*), putative glutamine transport system (*glnQ*, *glnM*, *glnH*), polar amino acid transport system (*yqiX*, *yqiY*, *yqiZ*), glycerol uptake facilitator (*glpF*, *glpP*), and iron complex transport system (*feuA*, *feuB*, *feuC*, *fhuB*) were downregulated.

#### 3.3.5. Gene Transcription and Regulation

There are numerous transcriptional regulation mechanisms in *B.amyloliquefaciens* that are used to regulate the synthesis of fengycin. RNA-seq revealed 43 DEGs associated with transcriptional regulatory factors, such as HTH-type transcriptional regulator (alsR, *ydhC2*, *citR*, *yuxN*, *cueR*, *MUS_4114*, *norG1*, *yazB1*), DeoR family transcriptional regulator (*fruR*), response regulator (*ComA*), the expression of 23 transcriptional regulatory factors, including polymerase sporulation-specific sigma factor (*SigH*, *SigW*) and two-component system (*degQ*, *degU*) were upregulated, while RNA polymerase sigma factor (*SigB*, *SigE*, *SigF*, *SigG*, *SigL*), transcriptional regulator (*AbrB*), stage 0 sporulation protein A (*Spo0A*), two-component response regulator (*PhrC*) and 19 other transcription factors were down-regulated. Based on transcriptome data and the regulatory mechanisms involved in the synthesis of lipopeptides in the previous study, we employed qRT-PCR to confirm the transcriptional regulatory factors: *ComA*, *SigH*, *AbrB*, *degQ*, *degU*, *PhrC*, *Spo0A*. As shown in Figure 7, the expression levels of *sigmaH* and *degU*-*degQ* two-component systems increased the most significantly.

## 4. Discussion

*B**. amyloliquefaciens* is a biological control potential strain that can produce fengycin and other surfactants with potent antifungal activities. In order to explain the gene expression pattern and compare the gene transcription under ML and MLF culture conditions, we screened out 1337 significantly differentially expressed genes focused on amino acid metabolism, fatty acid metabolism, energy metabolism, transport process, and gene transcription regulation process.

Fengycin is a lipopeptide molecule with a circular decapeptide structure; amino acids are its essential components. The lipopeptide synthesis process starts with connecting amino acids in a circular polypeptide structure under the action of NRPS synthetase and then combining with fatty acids to form an amphiphilic molecule [17]. Our previous research demonstrated the promoted production of fengycin by adding fructose to the medium [18]. The transcriptome results illustrated that adding fructose increased the biosynthesis of valine, leucine, isoleucine, and serine biosynthesis. For example, excess pyruvate produced by glycolysis is used as a precursor for amino acid synthesis. Pyruvate is acetylated during the synthesis of valine and alanine. Lactic acid synthase is converted into hydroxyethyl in the synthesis pathway of valine and isoleucine [19]. Fructose activates the expression of genes related to amino acid synthesis and improves the biosynthesis of fengycin.

As non-polar components of fengycin, the content of fatty acids is one of the limiting factors of lipopeptide synthesis. Fengycin production of *B.*
*subtilis* was increased to 130.11 mg/L after strengthening fatty acid metabolism [20]. The biological activity and yield of inturin and fengycin are increased in *B**. amyloliquefaciens* PC3 by adding alkanoic acid to the medium [21]. Fengycin synthesis is an energy-consuming process, and the adenylation of amino acids in the NRPS system requires the consumption of ATP [22]. Therefore, maintaining the dynamic balance of energy metabolism in cells is crucial for the synthesis of fengycin [23]. Transcriptome data show that the expression level of *yqjM* is increased by 4.3-times. Cytochrome c oxidase can discharge protons into the membrane gap, transfer electrons to oxygen, and produce water, and its expression drops to 0.47-times. In the presence of fructose, the energy-related genes of *B. amyloliquefaciens* fmb-60 alter, potentially affecting fengycin synthesis.

The addition of fructose influences the transport process of *B.*
*amyloliquefaciens*, which increases the expression of transporters of dominant carbon sources, such as fructose and glucose. In contrast, the expression of non-dominant carbon sources, such as lactose, ribose, and cellobiose transporters, was decreased due to carbon catabolite repression (CCR). When dominant carbon sources are present, bacteria preferentially utilize dominant carbon sources and suppress other carbon source metabolism and transport-related genes, reducing unnecessary gene expression and enhancing metabolic efficiency [24]. We discovered that the expression of all iron-transport-related genes in the transcriptome data were down-regulated, which could be related to the concentration of iron ions in the medium [25].

Fengycin synthesis is not affected by the gene transcription process but is regulated by the gene regulatory network because fengycin is synthesized by substrate catalysis. Lipopeptide synthesis is regulated by transcriptional factors (*sigmaH*, *Spo0A*, *AbrB*), two-component systems (*degU*, *degQ*), and quorum-sensing systems (*ComA*, *Phr**C*) [26]. As a transcription regulator, SigmaH recognizes the start site of mRNA transcription to promote gene expression [27]. The signal protein AbrB is a lipopeptide transcription inhibitor, and Spo0A is a transcription regulator in the sporulation process. Here, phosphorylated Spo0A plays a constructive role in fengycin production via inhibiting the expression of AbrB [28]. Recent studies have proved that DegQ increases the production of Fengycin [29]. Moreover, there are regulatory sites for ComA and DegU in the upstream region of the DegQ promoter, meaning that the expression of DegQ is regulated by ComA and DegU [30]. ComA is a response regulator in the quorum-sensing system during the formation of *Bacillus* competence. When the concentration of ComX reaches a certain threshold, ComA is phosphorylated to ComA-P by ComP and combined with the promoter of fengycin synthase to activate transcription [31]. PhrC is a competence, and sporulation factor (CSF) initiates the transcription by PhrC, binds to the phosphatase RapC, and inhibits its activity. Thus, reducing the impact of RapC on ComA-P, the dephosphorylation of this compound maintains the activity of ComA-P to ensure the transcription level of lipopeptide synthase. For instance, the production of surfactin in *B.subtilis* was increased by overexpressing the signal peptides PhrC and ComX [32]. Overall, the expression of *SigmaH*, *Spo0A*, *degU*, *degQ*, *ComA*, and *PhrC* are advantageous to transcription of the fengycin synthase gene and the fengycin production.

## 5. Conclusions

This study systematically analyzed gene expression about fengycin biosynthesis, amino acid metabolism, fatty acid metabolism, energy metabolism, and other critical factors, under different fructose supply conditions. Our analysis found 458 genes were upregulated and 879 genes were downregulated. Furthermore, the positive regulators of fengycin synthesis, *degQ*, *degU*, and *sigmaH,* were upregulated, while the negative regulator of *AbrB* was downregulated. In conclusion, this article describes the effect of fructose addition on fengycin synthesis by *B. amyloliquefaciens* at the transcriptional level and provides evidence for elucidating the regulation of fengycin synthesis. Our study shows changes in gene expression, but the mechanism of genes induce fengycin synthesis requires further investigation.

## Figures and Tables

**Figure 1 genes-13-00984-f001:**
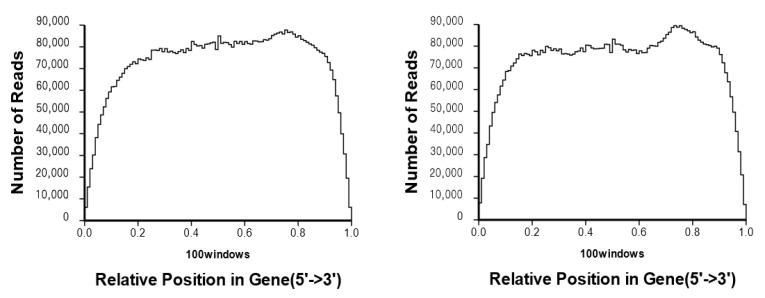
The distribution of transcriptome sequencing reads on the genome.

**Figure 2 genes-13-00984-f002:**
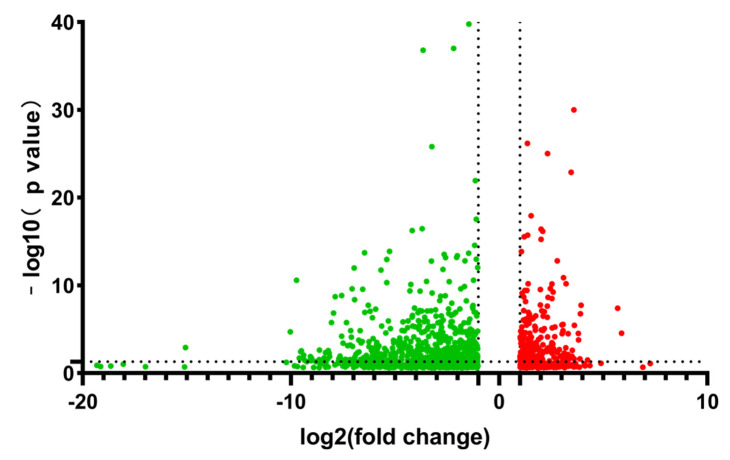
Volcano plot of *B**. amyloliquefaciens* differential gene expression under fructose induction.

**Figure 3 genes-13-00984-f003:**
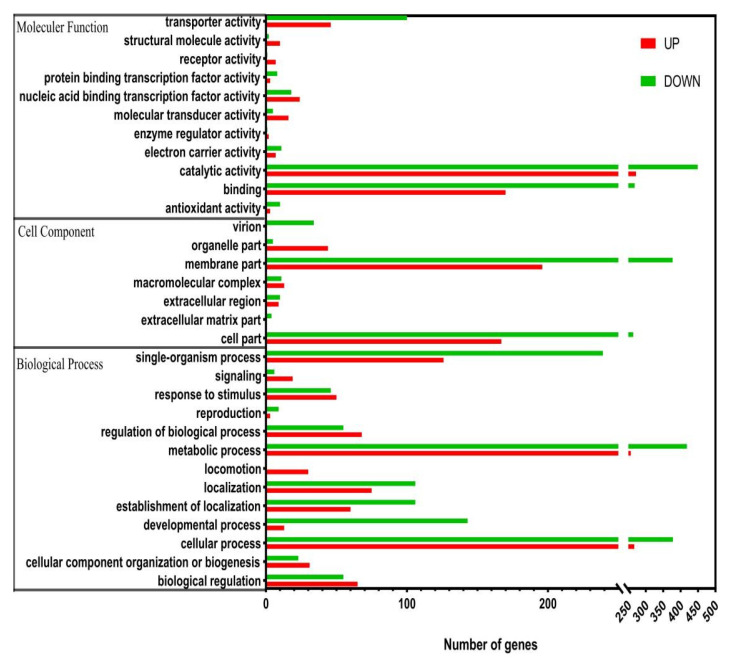
Gene ontology (GO) functional analysis of unique sequences from fructose and glucose transcriptome. Unique sequences were assigned to three categories: molecular function, cellular components and biological process.

**Figure 4 genes-13-00984-f004:**
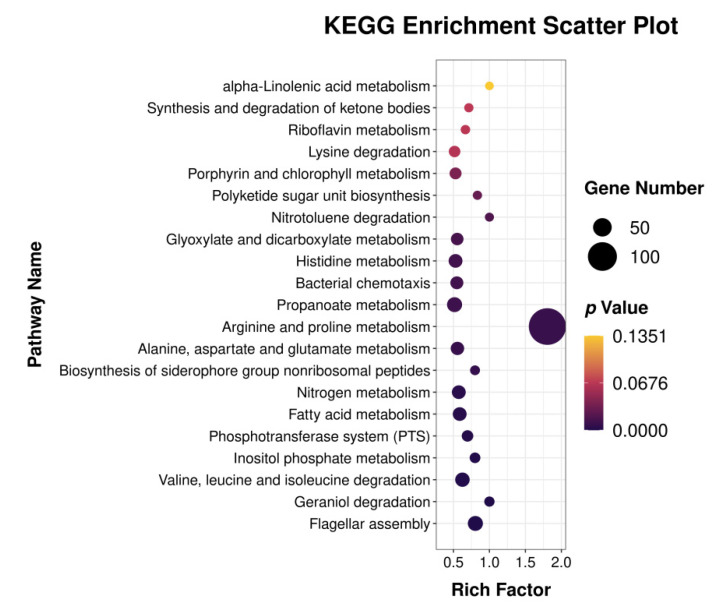
KEGG enrichment factor bubble chart.

**Figure 5 genes-13-00984-f005:**
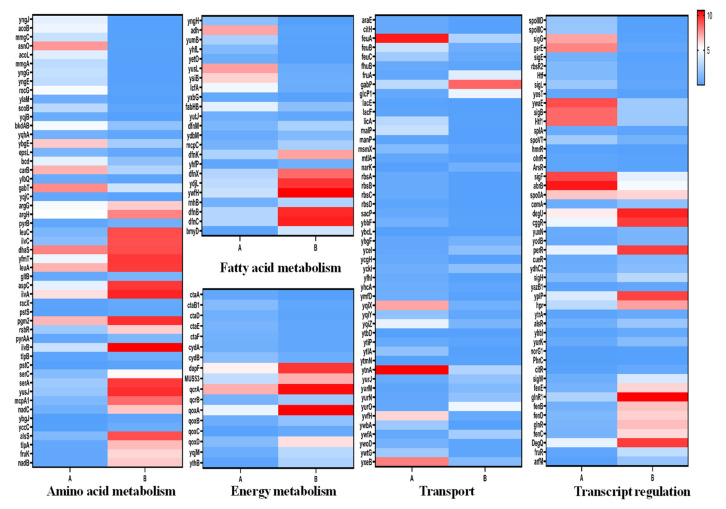
Gene expression change heat map, A: gene expression of *Bacillus amyloliquefaciens* cultured in ML medium, B: gene expression of *Bacillus amyloliquefaciens* cultured in MLF medium.

**Figure 6 genes-13-00984-f006:**
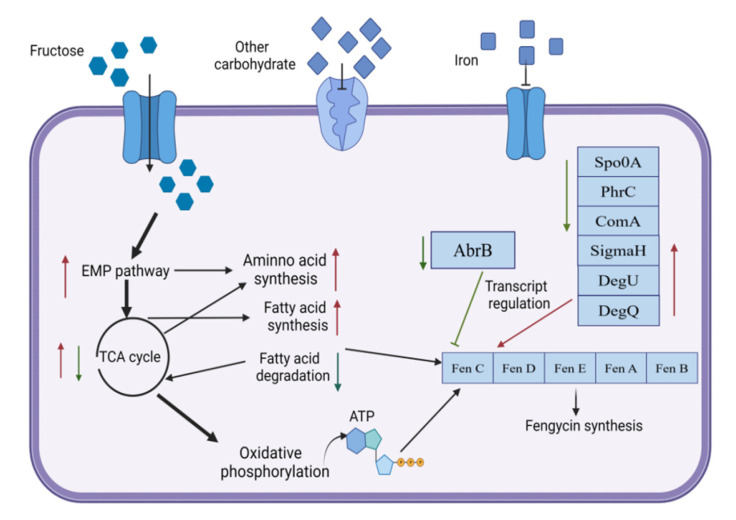
The effect of fructose on the synthesis of fengycin by *B. amyloliquefaciens*.

**Figure 7 genes-13-00984-f007:**
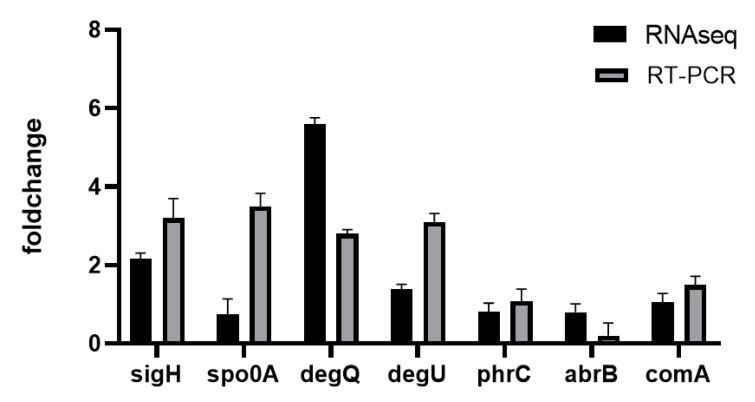
qRT-PCR analysis of fructose effects on the signal factors gene expression in *B**. amyloliquefaciens* fmb-60.

**Table 1 genes-13-00984-t001:** qRT-PCR primers used in this study.

Gene	Primer Name	Sequence5′-3′
*16s rRNA*	16s rRNA-FP	ACGGTCGCAAGACTGAAACT
	16s rRNA-RP	TAAGGTTCTTCGCGTTGCTT
*Sigma H*	Sigma H-FP	TTCAGGAAGGCATGATAGGC
	Sigma H-RP	GTGTTTCTGGCGAGTAGCTGT
*ComA*	ComA-FP	GCTCCATCCCATTGACCTC
	ComA-RP	TTGTCTGTTGATTGTCTCAGTCC
*degU*	degU-FP	GCAGAAACTCCGCTTGTTG
	degU-RP	GCTGAAAGAGATGGATGCTGAT
*AbrB*	AbrB-FP	TGGCAAGTCATGTTTGGTTT
	AbrB-RP	CGAACTGCGTCGTACTCTTG
*PhrC*	PhrC-FP	CAGCCGCGATTTTTACAGC
	PhrC-RP	CGTCATTCCTCTTTCTGTCACAT
*Spo0A*	Spo0A-FP	CAACGAGGAAATGGAATCAA
	Spo0A-RP	GCGAAGCAATCTCAATGGTAT
*degQ*	degQ-FP	ATGGTGAACGAGTCCTAGGT
	degQ-RP	TAGTCCTGTTCGCCAAATGC

**Table 2 genes-13-00984-t002:** Sequencing reads coverage ratio table.

Map to Genome	ML Reads Number	MLF Reads Number	ML Percentage	MLF Percentage
Total Reads	13303694	13586004	100.00%	100.00%
Total BasePairs	1197332460	1222740360	100.00%	100.00%
Total Mapped Reads	11681094	11907481	87.80%	87.65%
perfect match	3893131	4298706	29.26%	31.64%
≤5 bp mismatch	7787963	7608775	58.54%	56.00%
unique match	10813177	10378024	81.28%	76.39%
multi-position match	867917	1529457	6.52%	11.26%
Total Unmapped Reads	1622600	1678523	12.20%	12.35%

## Data Availability

The RNA-seq data have been submitted to the National Center for Biotechnology Information Sequence ReadArchive (http://www.ncbi.nlm.nih.gov/sra, accessed on 22 May 2022) under BioProject accession numbers PRJNA841238.

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
