# Peer review of "Transcriptome Analysis of Bacillus amyloliquefaciens Reveals Fructose Addition Effects on Fengycin Synthesis"

_genes, 2022, doi:10.3390/genes13060984_

Round 1
Reviewer 1 Report
I have some critical comments to presented paper.
- rpm value should be changed to " x g"
- in the discussion section, please add strengths and limitations of this study.
- in the abstract and in the main text, "conclusion" should be corrected and correspond with the results.
- in the introduction section, please clearly describe what it is known in this topic, and a gap - why this work is so important.
- abbreviations have to be explained when they were used at the first time.
- please add accession date of each database - 2.4. and describe this part in more details.
- please add catalog number and information about the companies for each reagent, software, equipment used in this study.
- some typo mistake should be eliminated.
- in general, the number of references are insufficient to support authors' observation.
Author Response
.

Reviewer 2 Report
The article examines the effect of fructose on the production of the antibiotic substance fengycin by amyloliquefaciens bacillus. The findings are based solely on the measurement of differential gene expression under fructose culture conditions. In order to have a more scientifically sound and be able to correctly interpret the results of the transcriptome analysis, it is necessary to perform the following:
1. The authors have to conduct physiological experiments to determine whether the reasons for the higher production of fengycin in fructose substrate are not due to higher cell density, better growth kinetics, or some other similar factor.
2. Genes affected by differences in gene expression can be represented in a Venn diagram, which will make their participation in fengycin production clearer.
3. Why are genes for the signaling factors selected for qRT-PCR analysis of fructose effects and not fenB, fenC, fenD, and fenE? I am a bit confused about the low level of overexpression of the last three genes. Twice the expression is not very high and needs additional supporting data, given that the +/- 1 overexpression or downregulation are insignificant values.
4. In my opinion, the table in the appendix is ​​redundant and these results should be presented in a figure that more clearly proves the genes sought by the authors involved in fengycin synthesis.
5. Reference [17] seems to be wrongly mentioned in the discussion. All references need revision to be in accordance with the journal's requirements. Please, add links.
6. The manuscript is full of syntactic and technical errors, there is even an error in the title. Authors are kindly requested to review the text very carefully before the second submission.
Author Response
.

Reviewer 3 Report
Manuscript "Analysis of Bacillus amyloliquefaciens transcriptome reveals fructose addtion effects on fengycin synthesis" is very interesting.
Authors analyzed the transcriptome of B. amyloliquefaciens fmb-60 under fructose culture conditions compared to glucose culture conditions, the results offer gene expression information to further research fengcyin metabolic pathways.
Results are very important but unfortunately lack of statistical analysis of obtainted data.
Line 154: "significantly changed" - On what basis?
Paper needs major revision.
Author Response
.

Reviewer 4 Report
General Comments:
This Reviewer agrees that the induction of fengycin synthesis in response to fucose is an interesting and significant finding; likewise the perturbation of the transcriptome shown by RNAseq and RT-qPCR in response to fucose addition is interesting. However, the mechanistic connection between the effects of fucose on the transcriptome and the regulation of fengycin synthesis remain somewhat unclear. Will any lipopeptide be induced in the presence of fucose? This would seem plausible given the hypothesis that amino acid synthesis, fatty acid availability, and transport increase in response to fucose. Could the Authors comment on whether the effect on fengycin synthesis is specific or more of a general effect on synthesis of all lipopeptides (and potentially other classes of natural products) in B. amyloliquefaciens?
Specific Comments:
Lines 48-49: “The genes accA (italics)…” etc.
Line 56: “…and the production of fengcyin is enhanced to 392.87 mg/L”
Methods section in general could use a careful re-read and correction of grammar and sentence syntax, including but not limited to:
Lines 79-84: Confusing, please consider re-wording this methods section.
Line 86: “analysis of genes was performed using Blast2Go…”
Line 87: “Transcripts were annotated by Blast…”
Lines 89-94: Confusing, please consider re-wording these sentences.
Results section:
Line 108: “The total number of sequenced bases was…”. NB: A “number of something” is considered singular, even though the number itself is greater than one. Same thing with line 111: This should read “the number of reads was counted”; or, you could say “reads were counted” (since “reads” is plural while “number of reads” is singular).
Lines 137-138: “Figure 2. Volcano plot of B. amyloliquefaciens differential gene expression under fructose induction.” or something along these lines.
Line 173: “enzymes is significantly increased…”; “improved” is a qualitative adjective.
Line 190: “…are up-regulated.”
Line 195-6: Possibly what is meant is “The transcriptional analysis revealed that 81 DEGs associated with transport systems were differentially regulated under fructose addition, including…”
Line 208-9: Possibly “There are numerous transcriptional regulation mechanisms in B.amyloliquefaciens that are used to regulate the synthesis of fengycin.”
Line 212: “response regulator (ComA).”
Line 244: “fatty acid content…”. Same with line 254.
Line 255: “Fengycin synthesis is an energy-consuming process, and the…”
Author Response
.

Round 2
Reviewer 1 Report
Some required parts by the Journal are missing (Institutional Review Board etc.)
Reviewer 2 Report
I think the text of the paper is corrected enough to be suitable for publication.
Reviewer 3 Report
Now, all is ok.